# Involvement of Vasopressin in Tissue Hypoperfusion during Cardiogenic Shock Complicating Acute Myocardial Infarction in Rats

**DOI:** 10.3390/ijms24021325

**Published:** 2023-01-10

**Authors:** Philippe Gaudard, Hélène David, Patrice Bideaux, Pierre Sicard, Jean-Paul Cristol, Gilles Guillon, Sylvain Richard, Pascal Colson, Anne Virsolvy

**Affiliations:** 1PhyMedExp, Université Montpellier, INSERM, CNRS, 34295 Montpellier, France; 2CHU de Montpellier, 34090 Montpellier, France; 3Institut de Génomique Fonctionnelle, Université Montpellier, INSERM, CNRS, 34094 Montpellier, France

**Keywords:** myocardial infarction, cardiogenic shock, vasopressin, copeptin, microcirculation

## Abstract

Acute heart failure (AHF) due to acute myocardial infarction (AMI) is likely to involve cardiogenic shock (CS), with neuro-hormonal activation. A relationship between AHF, CS and vasopressin response is suspected. This study aimed to investigate the implication of vasopressin on hemodynamic parameters and tissue perfusion at the early phase of CS complicating AMI. Experiments were performed on male Wistar rats submitted or not to left coronary artery ligation (AMI and Sham). Six groups were studied Sham and AMI treated or not with either a vasopressin antagonist SR-49059 (Sham-SR, AMI-SR) or agonist terlipressin (Sham-TLP, AMI-TLP). Animals were sacrificed one day after surgery (D_1_) and after hemodynamic parameters determination. Vascular responses to vasopressin were evaluated, ex vivo, on aorta. AHF was defined by a left ventricular ejection fraction below 40%. CS was defined by AHF plus tissue hypoperfusion evidenced by elevated serum lactate level or low mesenteric oxygen saturation (SmO_2_) at D_1_. Mortality rates were 40% in AMI, 0% in AMI-SR and 33% in AMI-TLP. Immediately after surgery, a sharp decrease in SmO_2_ was observed in all groups. At D_1_, SmO_2_ recovered in Sham and in SR-treated animals while it remained low in AMI and further decreased in TLP-treated groups. The incidence of CS among AHF animals was 72% in AMI or AMI-TLP while it was reduced to 25% in AMI-SR. Plasma copeptin level was increased by AMI. Maximal contractile response to vasopressin was decreased in AMI (32%) as in TLP- and SR- treated groups regardless of ligation. Increased vasopressin secretion occurring in the early phase of AMI may be responsible of mesenteric hypoperfusion resulting in tissue hypoxia. Treatment with a vasopressin antagonist enhanced mesenteric perfusion and improve survival. This could be an interesting therapeutic strategy to prevent progression to cardiogenic shock.

## 1. Introduction

Acute myocardial infarction (AMI) remains the main cause of heart failure and represents a major cause of death worldwide [1,2]. According to the European Society of Cardiology guidelines, left ventricular ejection fraction (LVEF) of 40% is the threshold defining heart failure with reduced ejection fraction [3]. Occlusion of the main coronary arteries results in an acute reduction of LVEF likely to lead to cardiogenic shock (CS). In humans, CS occurs in 5 to 10% of acute coronary syndromes with ST-segment elevation but is responsible for a mortality of 40 to 50% and requires high medical resources in intensive care units [4,5]. CS is defined by end-organ hypoperfusion due to primary cardiac dysfunction [3] and peripheral vasoconstriction. Diagnosis criteria in humans include persistent hypotension with cardiac index less than 2.2 L/min/m^2^, despite adequate or elevated filling pressure, and symptoms of end-organ hypoperfusion [1,6]. The course of cardiogenic shock usually includes two entangled hemodynamic profiles. On the one hand, the acute reduction of cardiac output (CO) induces vasoconstrictive response involving sympathetic system and vasoactive hormones (angiotensin II, vasopressin), attempting to maintain the perfusion pressure of organs. On the other hand, AMI and tissue hypoperfusion may lead to a systemic inflammatory response syndrome and an increased expression of inducible NO synthase, which are both responsible for inadequate vasodilation [1]. Therapeutic strategies aim to restore circulation and tissue perfusion. Thus, in addition to myocardial revascularization, pharmacological treatments consisting of vasopressors and/or inotropes are administered to support coronary and organ perfusion [4]. Norepinephrine is recommended as first-line therapy to restore quickly the blood pressure and possibly with the addition of an inotropic agent if stroke volume is not adequately improved [7]. Vasopressin is another agent utilized as second-line therapy for vasoconstriction. It is increasingly used in distributive shock and vasoplegia following cardiac surgery but not preferred in CS due to its lack of beneficial effect on cardiac output [7].

Vasopressin or arginine-vasopressin (AVP) is a nonapeptide synthesized in the hypothalamus from a prohormone, pre-pro-vasopressin, along with two other peptides: neurophysin II and copeptin, stored in the posterior pituitary and released after stimulation of baroreceptors. The hemodynamic action of AVP is mediated by the V_1a_ receptor in the vascular smooth muscle inducing vasoconstriction. With a better stability and a longer plasmatic half-life, copeptin can be used as a biomarker of AVP release [8,9]. Copeptin level at the hospital admission for AMI has a good predictive value for progression to chronic heart failure and one-year mortality [9,10]. Copeptin is a diagnostic and prognostic biomarker of cardiovascular diseases, including the rapid detection of acute myocardial infarction (AMI) and stroke, and the prediction of mortality in heart failure (HF) [11]. Elevated copeptin levels traducing vasopressin level increase are reported in the setting of AMI and shock [12]. This increase is thought to improve perfusion pressure at the cost of increased cardiac afterload, which may further impair myocardial function and promote ventriculoarterial uncoupling [1,13].

The model of choice for AMI in rats consists of the surgical left coronary artery ligation (LCA). This experimental model is interesting for monitoring myocardial remodelling, in particular the hemodynamic and morphological modifications which are similar to human clinical situations of acute coronary syndromes [14,15,16,17]. Although infarcts ranging from 4% to 65% have been described in various rat models [14,18], none mentions the use of this LCA ligation model to study acute hemodynamic response and CS.

The twofold objective of the present study was, on the one hand, to develop a rat model of CS following LCA ligation and, on the other hand, to investigate the implication of vasopressin system on acute hemodynamic changes and vascular reactivity during CS complicating AMI in that model. Thus, using a selective agonist, terlipressin, and a selective antagonist, SR-49059, of V_1a_ receptors, we showed that in the early phase of CS, vasopressin is involved in mesenteric tissue hypoperfusion. Additionally, treatment with SR-49059 significantly reduced animal mortality after LCA ligation.

## 2. Results

### 2.1. Groups Determination and Mortality Rates

The study included 103 rats spread out into 6 groups (Figure 1), depending on AMI and on treatment with either a vasopressin receptor antagonist (SR-49059, SR) or agonist (terlipressin, TLP).

Among rats with LCA ligation, 22 over 59 died at D_1_ after surgery in AMI without preoperative treatment (37.3%) but none in AMI-SR (*p* = 0.035). Thus, we observed that treatment with SR significantly reduced AMI-induced mortality. 

According to our definition criteria of AHF (LVEF < 40%), those with LVEF > 40% after LCA ligation at D_1_ were therefore excluded from the study. Among the 60 animals finally included, 16 were preoperatively treated with SR-49059 and 44 received no preoperative treatment. Thus, this defined two groups that we compared to evaluate the intrinsic effects of SR (Table 1). As shown in Table 1, no difference was observed regardless the examined parameter whether it be weight, heart rate, cardiac function or mesenteric oxygenation. Thus, no specific preoperative effect of SR was detected. 

### 2.2. Hemodynamic Changes Induced by Myocardial Infarction

We observed that myocardial infarction size did not differ between all AMI groups validating ligation and reproducibility of surgical procedure (Table 2). The average value of 45 ± 1% reflects a large left ventricular infarction. 

Cardiac and hemodynamic parameters at D_0_, before surgery, were not different between groups (Table 2). At D_1_, differences were observed for left ventricular end-diastolic diameter (LVEDD), left ventricular ejection fraction (LVEF), cardiac output (CO) and mean arterial pressure (MAP). These differences, predominantly significant between Sham and AMI whatever the treatment, are consistent with coronary ligation and myocardial ischemia. Averaged and compared to Sham groups, all AMI groups had higher LVEDD (7.4 ± 0.2 vs. 6.5 ± 0.1 mm; *p* = 0.0008), lower CO (109 ± 7 vs. 137 ± 7 mL/min; *p* = 0.007). Concerning LVEF, the reduced value observed in all AMI groups reflected our selection criteria, i.e., exclusion of animals with LVEF > 40% after ligation. However, if AMI induced a huge decrease of LVEF, we observed no difference between treatments. Concerning MAP, a lowering was observed in all AMI groups versus Sham groups (91 ± 5 vs. 123 ± 7, *t*-test: *p* < 0.001). When compared to AMI group, no change of MAP was demonstrated in AMI-SR (*p* = 0.251), while a hypertensive effect of terlipressin is highlighted (*p* = 0.003). Of note, a slight hypotensive tendency of SR was observed in Sham-SR compared to Sham (101 ± 6 vs. 118 ± 6, *t*-test: *p* = 0.101, ANOVA: *p* = 0.422).

### 2.3. Assessment of Tissue Perfusion

SmO_2_ under anesthesia at D_0_ before surgery was identical in all groups with an average value of 39.0 ± 0.6% (*p* = 0.368). Immediately after surgery, we observed a similar drop in SmO_2_ in all groups (*p* = 0.141) (Figure 2a), with an average value of 30.0 ± 0.5%, different from D_0_ value before surgery (*p* < 0.0001). The drop tended to be more severe in averaged AMI groups compared to Sham groups (−24.9 ± 1.9% vs. −19.1 ± 2.4%, *t*-test: *p* = 0.065). At D_1_, SmO_2_ increased and recovered in Sham, Sham-SR and AMI-SR while it remained low in AMI and in TLP-treated animals (Figure 2a). SmO_2_ values were lower in AMI compared to Sham (*p* = 0.032), in Sham-TLP compared to Sham (*p* = 0.005) and to Sham-SR (*p* = 0.009), and in AMI-TLP compared to AMI-SR (*p* = 0.003), (Figure 2b). No difference in SmO_2_ was noticed between Sham-SR and AMI-SR, with values in these groups similar to that of Sham.

Blood lactate levels are frequently used to assess impaired tissue oxygenation. Here, the highest plasma lactate levels were measured in AMI, AMI-TLP and also in Sham-TLP, with mean values greater than that of Sham (Figure 2c). In Sham-SR and in AMI-SR a moderate but not significant elevation of lactate level was noticed.

Both high lactate level and low SmO_2_ determined tissue hypoperfusion. A blood lactate level higher than 2.2 mmol/L indicated anaerobic metabolism and was therefore a marker of tissue hypoperfusion. Similarly, mesenteric hypoxia was evidenced when SmO_2_ was <35%. The incidence of mesenteric hypoxia was reduced in AMI-SR compared to AMI and to AMI-TLP (respectively 13%, 90% and 80%, *p* = 0.003). Combining those two parameters, we observed on Figure 2d that the incidence of tissue hypoperfusion was notably different between all groups (*p* = 0.001). Thus, tissue hypoperfusion was evidenced in nine AMI (69%), two AMI-SR (25%) and four AMI-TLP (80%) indicating a potential protecting effect of SR after AMI. Additionally, AMI did not increase the incidence of tissue hypoperfusion compared to Sham in SR-treated animals. Tissue hypoperfusion was also detected in Sham animals, with different incidence between groups, four Sham (19%), two Sham-SR (25%) and five Sham-TLP (100%) confirming the deleterious effect of TLP in animals without LCA ligation. 

### 2.4. Vasopressin System Stimulation: Copeptin

Copeptin is considered as a reliable surrogate marker of AVP secretion. Thus, in order to evaluate AVP system stimulation, we assayed copeptin in plasma samples. Under physiological conditions, vasopressin and copeptin circulating levels are low as confirmed by the copeptin concentration of 40 ± 6 pmol/L measured in control non operated rats (*n* = 3,). In all operated Sham animals, we observed plasma copeptin concentrations at D_1_ higher than the physiological basal value of controls, with an average value of 361 ± 42 pmol/L (Figure 3). No difference was detected in Sham between treated or not treated groups. Overall, copeptin level in AMI groups was higher than in Sham groups with significant differences between AMI and Sham (*p* = 0.034) and between AMI-SR and Sham-SR (*p* = 0.006). In AMI-SR, copeptin level even tended to be more important than in AMI (*p* = 0.057). In contrast, no difference was observed between Sham-TLP and AMI-TLP with a copeptin concentration in AMI-TLP lower than in AMI-SR (*p* = 0.002) and in AMI (*p* = 0.054). Of note, no increase of copeptin level was induced by AMI in TLP-treated animals.

### 2.5. Effect of Myocardial Infarction on Vascular Reactivity

To evaluate AVP sensitivity and efficiency in vascular bed, we analyzed the ex vivo contractile responses of rat aortic rings to cumulative doses of AVP normalized to the effect of a maximally active KCl concentration (60 mM). AVP induced concentration-dependent contractions in all groups (Figure 4). However, in AMI, the maximal contractile response and the sensitivity to AVP were decreased when compared to Sham with a 32% lowering of contraction (*p* = 0.011) and a slightly increased EC_50_ value (Table 3). In SR-treated animals, and similarly in Sham-SR and in AMI-SR, both contractile response and sensitivity were substantially decreased when compared to either Sham or AMI. In these groups, the strong increase in the EC_50_ values reflects the occupancy of the receptors by the competitive agonist SR 49059 [19]. Moreover, in TLP-treated animals, and similarly in Sham-TLP and in AMI-TLP, only a decrease of maximal contraction to AVP was observed, with no change of EC_50_ value. Of note, KCl-induced contraction and vasorelaxant effect of acetylcholine were identical in all groups. Only the response to PE differed with an increased contraction in TLP-treated animals either Sham or AMI (Table 3). 

Those results are in line with AVP receptors desensitization. In both SR- and TLP-treated animals this phenomenon is partially due to the respective presence of an antagonist and an agonist of AVP. Thus, in AMI, desensitization is to be linked to the increase in copeptin levels and therefore in AVP circulating levels.

## 3. Discussion

AMI with AHF represented a high-risk for CS (69% with tissue hypoperfusion) in the non-treated population. Increased plasma AVP levels have been observed in patients with AHF and have been shown predictive of severe HF development [20,21]. AVP could be involved in the onset of CS complicating AHF. In this study, we demonstrated for the first time that, in a rat model of CS, increased AVP blood level and vasopressinergic system activation contributed to mesenteric hypoperfusion. We showed that treatment with the AVP antagonist SR-49059 prevented tissue hypoperfusion and improved prognosis of CS while, in contrast, the agonist terlipressin aggravated them. These results suggested a critical role of vasopressinergic system in AMI-induced CS. 

Experimental myocardial infarction in rats induced by coronary artery ligation provides a clinically relevant model for the consequences of myocardial infarction. However, CS animal models are scarce and exploring CS in animal is challenging. We have developed such a model and defined the criteria that allowed assessing CS following coronary ligation. CS is generally defined as a state of end-organ hypoperfusion and hypoxia due to primary cardiac disorders. In humans, common CS criteria are reduced blood pressure and evidence of hypoperfusion assessed by end-organ dysfunction (neurologic disorder, oliguria, clod extremities, mottling) or elevated serum lactate (>2.2 mmol/L) [5]. In rats, no criteria are defined. In the study, we have set a LVEF threshold of 40% to define AHF after MI. With this selection criterion, we observed in AMI group, without vasopressin antagonist or agonist treatments, hypotension and reduced CO associated with higher lactate levels and lower SmO_2_ values. These data are consistent with some criteria of CS defined in human and allowed us to validate our model. 

Higher plasma copeptin levels were found in AMI. Copeptin is the C-terminal fragment of preprovasopressin which is co-secreted with AVP in the circulation. It is much more stable than AVP and could be used as a surrogate biomarker reflecting AVP production [9,22]. The increased copeptin level indicates a substantial vasopressin release in response to myocardial infarction or AHF in line with previous findings [20,21]. However, we noted that the plasma copeptin values were higher in the Sham groups (360 pmol/L) compared to the non-operated rats (40 pmol/L). This reflected the activation of the vasopressinergic system induced by the surgery, as indicated previously [23]. It is also known that, following an acute myocardial infarction, the arginine–vasopressin system as the renin–angiotensin and the sympathetic nervous systems, is activated [21,24,25,26,27]. We further observed an even higher copeptin level in AMI compared to Sham, revealing an additional release of AVP that can be interpreted as secondary to heart injury. Although such compensatory responses may be beneficial initially, epidemiological studies showed that sustained neurohumoral activation is associated with increased mortality [28]. Unlike the renin–angiotensin system and the sympathetic nervous system, the role of the vasopressin system in the development of heart failure is understudied. In our CS rat model, the activation of the vasopressin system is corroborated not only by the increase in AVP secretion but also by the decrease in the contractile response to AVP on isolated artery ex vivo. Indeed, in AMI group, we observed a lower Emax value and a slightly higher EC_50_ value which reflected a decreased sensitivity and efficiency of exogenous AVP consistent with receptor desensitization. This phenomenon, generally occurring after more or less prolonged exposure to the agonist [29], reveals here a prior vasopressinergic stimulation following the increase in AVP release. 

The effect of vasopressin on hemodynamic state and tissue perfusion during CS, was further investigated by treating animals either with a selective V_1a_ receptor antagonist, SR-49059, or with an agonist, terlipressin. SR-49059, with commercial name Relcovaptan^®^, is a potent and selective non-peptide vasopressin V_1a_ receptor antagonist devoid of agonist activity and displaying high affinity and efficacy for rat and human V_1a_ receptors [19]. Terlipressin is a vasopressin analogue with potent vasoconstrictive properties involving V_1a_ receptors [30]. To date, the effect of vasopressin receptor antagonists has not been described in the setting of CS. Several antagonists (vaptan) have been studied mainly focusing on chronic heart failure or on renal effect via V_2_ receptor inhibition [31,32]. In our study, the preoperative administration of SR-49059 had no incidence on the various measured parameters suggesting no effect on cardiac function and mesenteric oxygenation at baseline before surgery. At D_1_, a rather weak, decrease of MAP was observed in the Sham-SR group that could reflect a slight vasodilation with SR-49059 probably consecutive to an antagonism of the vasopressinergic system otherwise activated by surgery-induced stress [23]. This activation of the vasopressinergic system was also suggested by the high copeptin level compared to that of the non-operated animals. MAP was also decreased in AMI group. This hypotension at D_1_ in AMI group, despite a raise of copeptin level, is in agreement with the clinical observations and may be explained mainly by the decrease of CO insufficiently compensated by the vasoconstriction response. On the contrary, the increased MAP in animals treated with TLP is to be related to the high dose of agonist used, adapted to ensure a maximum vasoconstrictor effect. Since CO and systemic vascular resistance are the determinants of arterial blood pressure, a rise of MAP can be observed despite poor LV function and low CO in cases of major vasoconstriction. This hypertensive effect is also well documented [33]. It is known that shock state first induces an increase in AVP level causing vasoconstriction that helps to maintain end-organ perfusion. A fall in AVP level, probably due to depletion of stores, follows that initial increase and may contribute to hypotension and should be implicated in secondary vasoplegia and distributive shock [8,13,34]. Copeptin, due to its long half-life, could fail to follow rapid variations in AVP secretion and especially in decreases. This could certainly be a limitation in our model. Additionally, as shown by the lower copeptin level in TLP groups, treatment with terlipressin appears to have inhibited vasopressin release. The existence of such negative feedback through baroreceptors stimulation of the GABAergic pathway has been described from decades [35]. Furthermore, both agonist and antagonist treatments showed a competitive profile on aortic contractile responses to AVP consistent with the high affinity of SR-49059 and of terlipressin for the V_1a_ receptor [19,30].

An interesting observation is that treatment with SR-49059 had a positive impact on the evolution of AHF in AMI with improvement of tissue hypoperfusion and CS occurrence and outcome. In contrast, terlipressin treatment worsened the course of heart failure as shown by decreased SmO_2_ and increased serum lactate level in the TLP-treated groups. This is in line with a recently published study showing that ultra-low dose of exogenous AVP increased gastric microcirculatory oxygenation (µHbO2) while high dose of AVP reduced it [36]. The effects of AVP on µHbO2 were otherwise completely abolished by V_1a_ receptor inhibition. Our finding supports a deleterious effect related to endogenous AVP release following AMI, involving prolonged mesenteric ischemia and anaerobic metabolism. 

This is the first description of vasopressin implication in microcirculation alteration in the context of post-ischemic CS. These findings are consistent with a preliminary prospective human cohort of patients with CS complicating AMI [37]. In this short series, patients had both severe impairment of microvascular reactivity and high copeptin level and early reversal of microcirculation impairment in response to hemodynamic support was associated with survival. The potential clinical implications of these results are to assess vasopressinergic system stimulation by measuring the level of plasma copeptin in the first hours after AMI and to find a way to prevent excessive vasopressin release. The potential protective effect of antagonist treatment must be explored and confirmed in case of administration after the onset of coronary occlusion to be transposable for clinical use. In the context of quick reduction of maximal contractile response to AVP, the agonist treatment appears unsound to restore arterial pressure. In case of vasoplegic syndrome following AMI, therapeutic compensation of vasopressin relative deficiency should be considered only once arterial blood flow is well restored, microcirculation closely monitored, and this treatment is quickly eliminated if not tolerated.

In summary, these results highlight the critical role of vasopressinergic response in the very early phase of ischemic AHF and its potential impact on time course of CS. Indeed, the acute release of vasopressin participates in the vasoconstrictive response of CS which induces, as a salvage response, a redistribution of blood flow to autoregulated organs but which is also responsible for side effects in particular mesenteric ischemia. This tissue damage may lead to organ dysfunction and contribute to severe complications, such as bacteria translocation from the gut, and to the secondary systemic inflammatory response observed in CS [5]. 

## 4. Materials and Methods

### 4.1. Animal Ethics 

All experimental procedures were conducted in accordance with the European Union Laboratory Animal Care Rules (2010/63/EU Directive) and NIH Guidelines. The project (APAFIS#9808) was approved by the local committee for Animal Care of Montpellier-Languedoc-Roussillon (N° CEEA-LR-9808). Healthy male Wistar rats (350–400 g) were maintained in our animal facility one week before experiments with free access to food and water.

### 4.2. Experimental Design

Animals were randomly assigned for various surgical procedures and treatments. AMI was surgically induced by left coronary artery ligation. Fake surgery was performed in Sham. Six groups were studied: Sham and AMI without or with treatment with either terlipressin (Sham-TLP and AMI-TLP) or SR-49059 (Sham-SR and AMI-SR). Animals were sacrificed one day after surgery (D_1_). In order to study AMI-induced AHF, a LVEF ≤ 40% was a prerequisite for the selection of animals in AMI groups. To assess CS, a tissue hypoperfusion (lactate > 2.2 mmol/L) or mesenteric hypoxia (SmO_2_ < 35%) was required. 

### 4.3. Surgical Procedure

Animals were anaesthetized by intraperitoneal injection of ketamine 130 mg/kg and xylazine 8.5 mg/kg and intubated with a 14-gauge catheter. They were mechanically ventilated with 2.5 mL of tidal volume by 70 cycles per minute. The heart was exposed for ligation after a left para-sternal thoracotomy and incision of the pericardium. AMI was induced by permanent ligation (Appendix A, at 1 to 2 mm from the left atrial appendage line, of the LCA and the efficacy of the ligation was assessed by changes in electrocardiogram and in color of anterior myocardium turning white. At the end of surgery, pneumothorax was drained off, local analgesia was achieved with ropivacaine 2 mg/mL 0.5 mL/kg, skin was closed with Premio 5-0 stitches (Peters Surgical^®^) and 1 mL of isotonic saline solution was injected in peritoneum for volume compensation. Sham surgery was performed according to the same protocol except closing of the ligation. After surgery, rats were allowed to recover in individual cages. They were regularly checked and had free access to food and water. They were housed under conditions of controlled temperature (22–24 °C), humidity (45–50%) and light cycle (12 light hours/12 dark hours). One day after surgery (D_1_), full hemodynamic evaluation was performed under the same protocol of general anesthesia and animals were then sacrificed after exsanguination. 

Infarct size was determined after sacrifice and according to previously validated method [38]. The heart was quickly excised and sliced into four 2.0-mm-thick sections perpendicular to the long axis. Sections were incubated in 1% triphenyltetrazolium chloride at 37 °C for 10 min and then imaged using a colour flat-bed scanner at 600 dpi resolution (see illustration on Appendix A). Healthy myocardium appeared in red and dead tissues in white. The infarcted area, colored in white, was determined by computerized planimetry with ImageJ^®^ 1.52v software in each slice. The infarcted area/total LV myocardial area ratios for each slice were calculated. Results are expressed in average of percentage of infarcted area on total area on the 4 slices.

### 4.4. Heart Rate and Blood Pressure Measurement

The heart rate was determined by electrocardiography monitoring at D_0_ and D_1_ during anesthesia. The arterial pressure was measured invasively at D_1_ via cannulation of the right carotid artery using a pressure transducer connected to a monitor (ADInstruments, Sidney, Australia). The pressure signal was analyzed by a data acquisition software (LabChart Reader 8.1.5, ADInstruments, Sidney, Australia) and the mean arterial pressure (MAP) was determined.

### 4.5. Echocardiography

High resolution echocardiography (VisualSonics/Fujifilm, Toronto, ON, Canada with a MS250D ultrasound probe 20 MHz) was performed at D_0_ before surgery and at D_1_ before sacrifice, under anesthesia by 2% isofluorane inhalation at 37 °C, and under ECG and respiratory rate monitoring, as previously described [39]. Transthoracic echocardiographic recordings are illustrated in Appendix A. Left ventricular (LV) parasternal long axis 2D view in B-mode and M-Mode was performed at the level of papillary muscle to assess LV wall thicknesses and internal diameters, allowing the calculation of LVEF and left ventricular end diastolic diameter (LVEDD). The velocity time integral of blood flow in the ascending aorta was used to calculate CO. 

### 4.6. Mesenteric Haemoglobin Oxygen Saturation

Tissue oxygenation was assessed by a regional measurement of mesenteric haemoglobin oxygen saturation (SmO_2_) with a transcutaneous NIRS device (INVOS™ 5100C cerebral/somatic oximeter, Medtronic, Brampton, ON, Canada), reflecting the oxygen content in microcirculation (vessels diameter < 1 mm) through a sensor applied on the abdominal area between the xiphoid appendix and bladder, analyzing the tissue at a depth of 1.5 cm as illustrated in Appendix A. Determinations of SmO_2_ were performed at D_0_ just before and immediately after the surgery and at D_1_, by the average of the acquisitions obtained from two positions of the NIRS sensor. 

### 4.7. Lactate and Copeptin Assay

Blood sample collected in an EDTA tube (8 mL) was immediately centrifuged for 12 min at 1300× *g*, 4 °C and plasma was frozen at −80 °C for further assays. Lactate concentration was determined with Cobas 8000 analyzer (colorimetric assay, reference interval [0.5–2.2] mM, Roche Diagnostics, Meylan, France). Copeptin was assayed using ELISA kit (Cusabio Technology, Houston, TX, USA) according to manufacturer instructions. 

### 4.8. Vascular Reactivity

Experiments were performed as previously described [40] on isolated aortic rings mounted onto thin stainless-steel holders and placed in organ chambers filled with a saline solution (PSS) maintained at 37 °C and continuously bubbled with O_2_. Changes in isometric tension were measured using a force transducer (IT1-25) and recorded by an IOX computerized system (EMKA Technologies, Paris, France).

After an equilibration period of 1 h at a resting tension of 2 g, functional integrity of each arterial segment was evaluated by the contractile response to a single dose of phenylephrine (PE, 10 µM) and then by the relaxant effect of acetylcholine (ACh, 1 µM) which validated endothelium function. After a wash-out and stabilization period of 20 min, the contractile capacity of vascular smooth muscle cells was evaluated with cumulative dose responses to the different agonists: AVP, KCL, or PE. The relaxation was studied with cumulative doses of ACh on arterial rings previously contracted with a submaximal dose of PE (10 µM).

### 4.9. Drugs

SR-49059 (Relcovaptan) is a selective nonpeptidic V_1a_ receptor antagonist. It was obtained from Sanofi-Aventis (Sanofi-Aventis, Toulouse, France). SR-49059 was dissolved in DMSO (48 mg/mL) then extemporaneously emulsified in Cremophor^®^ RH40 (Sigma-Aldrich, Saint-Quentin-Fallavier, France) and diluted in 9% sodium chloride solution before injection. Treatment consisted of three intraperitoneal injections of SR-49059 solution performed, 1 h before anesthesia (4.5 mg/kg), 8 h after the surgery (6 mg/kg) and 1 h before sacrifice (4.5 mg/kg). 

Terlipressin is a well-documented vasopressin receptor agonist inducing intense and prolonged vasoconstriction. Commercial terlipressin, Glypressine^®^, was obtained from Ferring GmbH (Kiel, Germany). Its purity has been previously evaluated by mass spectrometry [30]. Terlipressin solution (20 µg/mL) was prepared by solubilization in 9% sodium chloride. Treatment consisted of three intraperitoneal injections of terlipressin solution performed 1, 12 and 23 h after surgery and corresponding to a total dose of 80 µg/kg/24 h. To reduce the risk of bleeding due to the increase of blood pressure, terlipressin was not administered either preoperatively or immediately after surgery.

### 4.10. Statistical Analysis

Results, expressed as mean and standard error of the mean (SEM), were compared between non treated groups (Sham and AMI), antagonist treated groups (Sham-SR and AMI-SR) and agonist treated groups (Sham-TP and AMI-TP) by an ordinary one-way ANOVA or a Kruskal–Wallis test according to the variable distribution. An appropriate post-hoc multiple comparisons test was used to determine which groups are different. Paired *t*-test or Wilcoxon test were used for comparisons between D_0_ and D_1_ within each group. Distribution of outcome endpoints were compared by a Fisher’s exact test. All probability values are 2-sided, and values of *p* < 0.05 were considered to be significant. Statistical analyses were performed using GraphPad Prism 6 (GraphPad Software, Inc., San Diego, CA, USA).

## 5. Conclusions

CS after AMI was characterized by a sustained decrease in mesenteric oxygenation and by lactate production. Increased vasopressin secretion seems involved in the early phase of AHF, which may result in worsening mesenteric hypoperfusion despite quick reduction in vascular reactivity to vasopressin. High vasopressin or agonist stimulation following AHF may explain the alteration of arterial contractile response to vasopressin and the implication of this mechanism in secondary vasodilation syndrome needs to be further investigated.

## Figures and Tables

**Figure 1 ijms-24-01325-f001:**
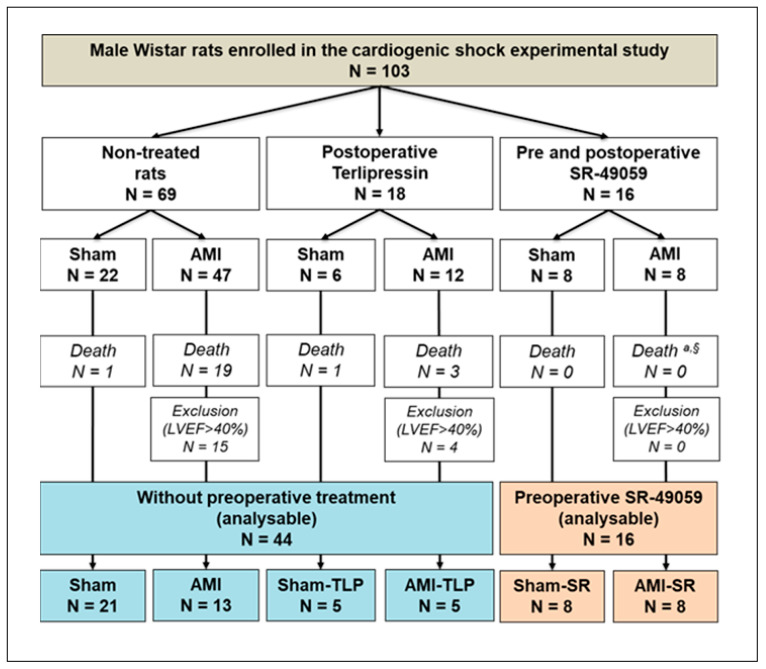
Experimental design and group selection for analysis at postoperative day 1 (D_1_). AMI, acute myocardial infarction; SR-49059, vasopressin receptor antagonist; LVEF, left ventricular ejection fraction; TLP, Terlipressin. ^a^ Fisher exact test between the three AMI groups: *p* = 0.057. ^§^ Chi-2 test between preoperatively non-treated AMI vs. SR-treated AMI groups: *p* = 0.035.

**Figure 2 ijms-24-01325-f002:**
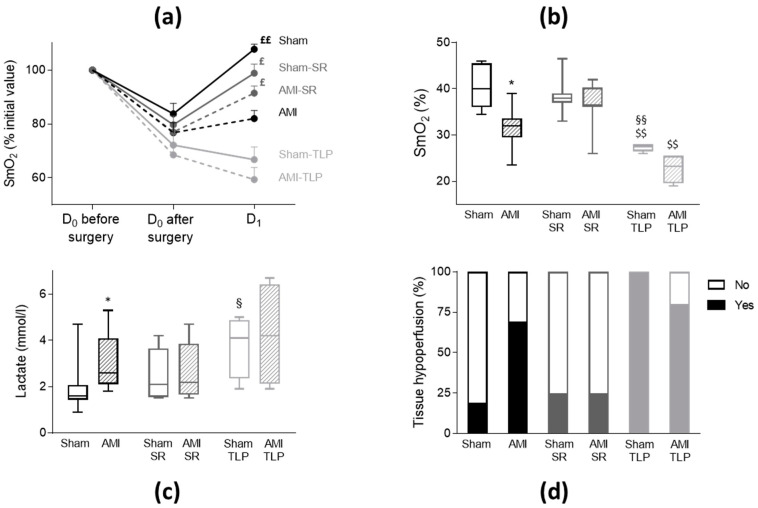
Tissue hypoperfusion assessment by mesenteric oxygenation saturation (SmO_2_) and lactate level. (**a**) Variation of SmO_2_ measured for all groups at D_0_ before surgery, at D_0_ immediately after surgery and at D_1_. Values were normalized to the initial value at D_0_ and data are mean ± SEM. ^£^
*p* < 0.05, ^££^
*p* < 0.01 for paired measures D_1_ vs. D_0_ after surgery in each group. (**b**) Box and whisker plots of SmO_2_ values at D_1_ (Kruskal–Wallis test, *p* < 0.0001). * *p* < 0.05 for AMI versus Sham with the same treatment. ^§§^
*p* < 0.01 for treated versus non-treated group for the same coronary intervention. ^$$^
*p* < 0.01 AMI-SR vs. AMI-TLP and Sham-SR vs. Sham-TLP. (**c**) Box and whisker plots of plasma lactate level at D_1_ (Kruskal–Wallis test, *p* = 0.034). * *p* < 0.05 for AMI versus Sham with the same treatment. ^§^
*p* < 0.05 for treated versus non-treated group for the same coronary intervention. (**d**) Distribution of animals with tissue hypoperfusion, defined as lactate > 2.2 mmol/L or SmO_2_ < 35% at D_1_, and expressed as percentage (Chi-square: *p* = 0.001). No: no hypoperfusion; Yes: with hypoperfusion.

**Figure 3 ijms-24-01325-f003:**
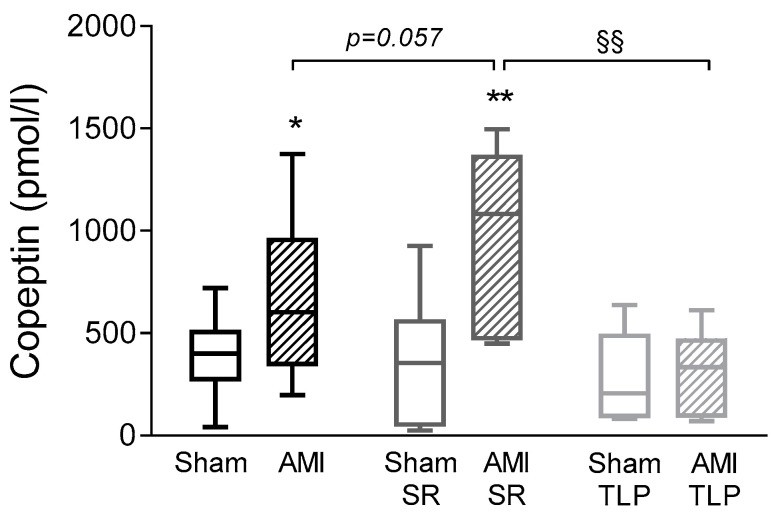
Copeptin levels measured at D_1_ before sacrifice in all groups. Data are presented as box and whisker plots. Analyses were performed using an ordinary one-way ANOVA (*p* = 0.001) followed by Holm–Sidak’s multiple comparisons test. * *p* < 0.05, ** *p* < 0.01 for AMI versus Sham with the same treatment; ^§§^
*p* < 0.01 for the effect of treatment in the same coronary intervention.

**Figure 4 ijms-24-01325-f004:**
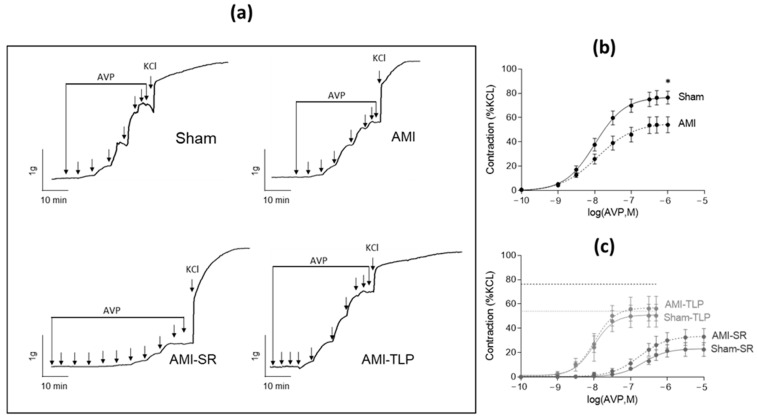
Vascular reactivity: ex vivo vasopressin-induced contractile responses of rat aortic rings. (**a**) Representative recordings of isometric tension variations induced by cumulative doses of AVP ranging from 0.1 nM to 1 or 10 µM followed by a maximally active dose of KCl (60 mM) for Sham, AMI, AMI-SR and AMI-TLP. (**b**,**c**) Dose–response curves to AVP normalized to the maximal response induced by KCl (60 mM) in the non-treated groups (**b**) and in the SR-49059 and Terlipressin treated groups (**c**). The thick and fine dashed lines indicate the maximal AVP-induced contraction in Sham and AMI, respectively. Data are mean ± SEM. * *p* < 0.05 for AMI versus Sham (Student’s *t*-test).

**Table 1 ijms-24-01325-t001:** Initial characteristics of animals after general anesthesia and before surgery without (non-treated and post-operative Terlipressin groups) and with pre-operative treatment with SR-49059 (administered 1 h before anesthesia).

Baseline Characteristics	Without Preoperative Treatment (*n* = 44)	SR-49059 Preoperative Treatment (*n* = 16)	*p*
Weight, g	369.1 ± 3.0	361.6 ± 3.6	0.174
Heart rate, bpm	270 ± 5	292 ± 19	0.431
LVEDD, mm	7.4 ± 0.1	7.1 ± 0.2	0.176
Cardiac output, mL/min	159.1 ± 9.2	143.1 ± 14.2	0.348
LVEF, %	65.5 ± 0.8	66.3 ± 2.0	0.648
SmO_2_ at baseline, %	38.7 ± 0.8	39.6 ± 0.8	0.368

Data are expressed as mean ± SEM, Student’s *t*-test or Mann-Whitney test used as appropriate for comparison between groups. SR-49059, vasopressin receptor antagonist; LVEF, left ventricular ejection fraction; bpm, beats per minute; LVEDD, left ventricular end-diastolic diameter; SmO_2_, mesenteric hemoglobin oxygen saturation (baseline is the first measurement at day 0 after general anesthesia and before surgery).

**Table 2 ijms-24-01325-t002:** Cardiac and hemodynamic parameters evaluated before (D_0_) and 1 day after surgery (D_1_) in all groups submitted or not to coronary artery ligation and treated or not with SR-49059 or Terlipressin.

Parameters	Sham (*n* = 21)	AMI (*n* = 13)	Sham-SR (*n* = 8)	AMI-SR (*n* = 8)	Sham-TLP (*n* = 5)	AMI-TLP (*n* = 5)	*p*
Weight D_0_, g Weight D_1_, g	369.6 ± 4.6 358.2 ± 3.9 ^a^	368.5 ± 4.9 354.9 ± 4.8 ^a^	360.6 ± 4.5 345.9 ± 4.7 ^a^	362.6 ± 5.9 346.7 ± 5.3 ^a^	380.2 ± 9.9 365.8 ± 9.8	357.2 ± 6.4 339.6 ± 7.0	0.429 0.132
HR D_0_, bpm HR D_1_, bpm	273 ± 7 306 ± 10 ^a^	263 ± 11 308 ± 11 ^a^	310 ± 36 299 ± 11	274 ± 14 333 ± 17	278 ± 26 312 ± 9	274 ± 12 345 ± 25	0.889 0.382
LVEDD D_0_, mm LVEDD D_1_, mm	7.4 ± 0.2 6.5 ± 0.1 ^a^	7.7 ± 0.1 7.6 ± 0.4 *	6.6 ± 0.2 6.3 ± 0.3	7.8 ± 0.2 7.2 ± 0.4	7.5 ± 0.5 6.8 ± 0.2	7.1 ± 0.3 7.2 ± 0.3	0.079 **0.038**
LVEF D_0_, % LVEF D_1_, %	66.1 ± 1.4 60.3 ± 1.7	64.6 ± 1.0 34.4 ± 1.3 ^a,^*	64.2 ± 2.5 58.5 ± 2.2	68.4 ± 2.9 35.9 ± 1.6 ^a,^*	67.0 64.7 ± 5.4	65.3 ± 3.0 33.2 ± 3.5 *	0.700 **<0.001**
CO D_0_, mL/min CO D_1_, mL/min	149.8 ± 9.0 144.1 ± 8.9	186.9 ± 19.7 112.6 ± 9.9 ^a^	140.0 ± 22.4 122.6 ± 15.3	146.2 ± 20.2 104.6 ± 8.3	145.5 ± 28.5 130.0 ± 19.3	113.0 ± 13.0 106.8 ± 21.3	0.225 0.082
Δ CO, %	0.1 ± 6.2	−28.1 ± 6.7 *	−12.2 ± 11.6	−15.6 ± 11.6	_ ^1^	_ ^1^	**0.048**
MAP D_1_, mmHg	118 ± 6	81 ± 4 *	101 ± 6	87 ± 7	160 ± 5	123 ± 4 ^§^	**<0.001**
MI size, % of LV		45.5 ± 2.7		44.6 ± 1.5		44.2 ± 2.9	0.812

Data are expressed as mean ± SEM, one-way ANOVA or Kruskal–Wallis test were used as appropriate to compare all groups with a multiple comparisons test to compare Sham vs. AMI in the same category of treatment and Sham or AMI non-treated groups vs. SR and TLP treated groups. AMI, acute myocardial infarction; SR, vasopressin receptor antagonist SR-49059; TLP, Terlipressin; LVEF, left ventricular ejection fraction; HR, heart rate; bpm, beats per minute; MAP, mean arterial pressure; CO, cardiac output; Δ CO, variation of CO from baseline (D_0_) expressed as percentage of D_0_ value. * *p* < 0.05 compared to Sham within the same treatment category (none, SR or TLP). ^§^
*p* < 0.05 compared to non-treated groups for the same coronary intervention (Sham or AMI). ^a^
*p* < 0.05 for paired measures D_1_ vs. D_0_. ^1^ missing paired values for variation calculation.

**Table 3 ijms-24-01325-t003:** Summary of ex vivo contractile response of aortic rings to various agonists vasopressin (AVP), potassium chloride (KCl) or phenylephrine (PE) and to the antagonist acetylcholine (ACh).

Parameters	Sham (*n* = 20)	AMI (*n* = 13)	Sham-SR (*n* = 8)	AMI-SR (*n* = 8)	Sham-TLP (*n* = 5)	AMI-TLP (*n* = 5)	*p*
AVP E_max_, (%KCl)	76.7 ± 5.7	51.8 ± 6.3 *	26.3 ± 6.5 ^§^	34.4 ± 7.7	50.4 ± 9.4 ^§^	56.2 ± 10.1	**<0.0001**
AVP EC_50_, nM	12 ± 2	59 ± 25	215 ± 50 ^§^	221 ± 45 ^§^	12 ± 3	11 ± 2	**<0.0001**
KCl E_max_, g	2.2 ± 0.2	2.7 ± 0.2	2.6 ± 0.2	3.2 ± 0.1	2.8 ± 0.3	2.6 ± 0.1	0.061
PE E_max_, g	2.4 ± 0.1	2.8 ± 0.2	3.2 ± 0.3	3.3 ± 0.2	3.7 ± 0.6 ^§^	3.9 ± 0.2 ^§^	**0.001**
ACh relax, (%PE)	68.3 ± 3.8	68.8 ± 3.4	75.7 ± 5.0	73.8 ± 5.3	62.1 ± 6.4	68.7 ± 3.6	0.439

Values represent mean ± SEM. EC_50_ values are determined from dose-response curves to AVP expressed as the percentage of maximal contraction induced by 60 mM of KCl. E_max_ for AVP was expressed as %of the maximal contraction induced by KCl (60 mM). Ordinary one-way ANOVA or Kruskal–Wallis test were performed to compare all groups. * *p* < 0.05 for AMI versus Sham within the same treatment category (none, SR or TLP). ^§^
*p* < 0.05for treated versus non-treated groups for the same coronary intervention (Sham or AMI). AVP, arginine vasopressin; KCl, potassium chloride; PE, phenylephrine; ACh, acetylcholine; AMI, acute myocardial infarction; SR, SR-49059; TLP, terlipressin.

## Data Availability

The data presented in this study are available on request from the corresponding author. The data are not publicly available due to privacy.

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
