# Peer review of "Involvement of Vasopressin in Tissue Hypoperfusion during Cardiogenic Shock Complicating Acute Myocardial Infarction in Rats"

_ijms, 2023, doi:10.3390/ijms24021325_

Round 1

Reviewer 1 Report

The authors developed a rat model of acute heart failure and cardiogenic shock and studied the effect of vasopressin system on hemodynamic parameters and tissue oxygenation. They showed that vasopressin is involved in mesenteric tissue hypoperfusion early after the development of AMI. Treatment with vasopressin antagonist had a positive effect on the studied parameters and reduced animal mortality.

Comments.

1.     The rather weak hypotonic effect of the vasopressin antagonist along with a pronounced rise in blood pressure in the presence of the agonist is noteworthy.  The authors could explain this phenomenon.

2.     In the caption to figure 1, the authors decode the abbreviation "SmO2, mesenteric haemoglobin oxygen saturation». However, this index is not shown in the figure.

3.     Paragraph 2.2, line 130 - the sentence is incomplete.

4.     There are few typos and errors, extra commas and spaces that should be corrected.

Author Response

  1. The rather weak hypotonic effect of the vasopressin antagonist along with a pronounced rise in blood pressure in the presence of the agonist is noteworthy.  The authors could explain this phenomenon.

Answer: Such precision has added in the discussion. We had only discussed the weak hypotonic effect of SR-49059. The hypertensive effect of vasopressin and vasopressin agonists like terlipressin are well known and documented (https://doi.org/10.1016/j.bpa.2008.02.006). However, we have now added a short explanation for the effect of terlipressin on MAP in TLP treated groups: “On the contrary, the increased MAP in animals treated with TLP is to be related to the high dose of agonist used, adapted to ensure a maximum vasoconstrictor effect. Since CO and systemic vascular resistance are the determinants of arterial blood pressure, a rise of MAP can be observed despite poor LV function and low CO in cases of major vasoconstriction.” (lines 324-328)

  1. In the caption to figure 1, the authors decode the abbreviation "SmO2, mesenteric haemoglobin oxygen saturation». However, this index is not shown in the figure.
  2. Paragraph 2.2, line 130 - the sentence is incomplete.
  3. There are few typos and errors, extra commas and spaces that should be corrected.

Answer: we thank the reviewer for its comments and the careful reviewing of the manuscript. The abbreviation has been suppressed (point 2), the incomplete sentence has been deleted (point 3) and we hope that the main typos, commas and spaces have been overall corrected (point 4).

Reviewer 2 Report

Dear colleagues,

I am more a specialist of vessels than heart. For me, the manuscript is clear and describes how the vasopressinergic system might be implicated in cardiogenic chock linked to acute myocardial infarction. I have a few remarks concerning your manuscript to improve the understanding of the conduct of experiments. Can you illustrate the measurement of infact? the description given in paragraph 4.3 seems too succinct to me.

The presentation of images in Figure S1 could be accompanied by a graph to improve understanding.

Is it possible to add an additional figure to better describe the ultrasound procedure?

minor points

l52 2.2 L/min/m(square) put the 2 as exponent

l130, there is a problem with the last sentence Myocardial infarction was evaluated : there is not point at the end of the sentence because it is forgotten ? is the sentence not finished or could be deleted ?

be careful with the abbreviation of Acetylcholine (ACH, Ach or ACh, the last should be the best).

Author Response

Comments and Suggestions for Authors

Dear colleagues,

I am more a specialist of vessels than heart. For me, the manuscript is clear and describes how the vasopressinergic system might be implicated in cardiogenic chock linked to acute myocardial infarction. I have a few remarks concerning your manuscript to improve the understanding of the conduct of experiments. Can you illustrate the measurement of infact? the description given in paragraph 4.3 seems too succinct to me.

The presentation of images in Figure S1 could be accompanied by a graph to improve understanding.

Is it possible to add an additional figure to better describe the ultrasound procedure?

Answer: We agree with the reviewer and modified the corresponding section in methods to explain as suggested how the infarct size was measured according to a referenced method (lines 406-417) (ref 38: doi:10.3791/54914): “Infarct size was determined after sacrifice and according to previously validated method [38]. The heart was quickly excised and sliced into four 2.0-mm-thick sections perpendicular to the long axis. Sections were incubated in 1% triphenyltetrazolium chloride at 37°C for 10 min and then imaged using a colour flat-bed scanner at 600 dpi resolution (see illustration on supplemental figure S1). Healthy myocardium appeared in red and dead tissues in white. The infarcted area, colored in white, was determined by computerized planimetry with ImageJ® software in each slice. The infarcted area/total LV myocardial area ratios for each slice were calculated. Results are expressed in average of percentage of infarcted area on total area on the 4 slices.”

minor points

l52 2.2 L/min/m(square) put the 2 as exponent –

Answer: Ok corrected

l130, there is a problem with the last sentence Myocardial infarction was evaluated : there is not point at the end of the sentence because it is forgotten ? is the sentence not finished or could be deleted ?

Answer: Ok this sentence has been suppressed.

be careful with the abbreviation of Acetylcholine (ACH, Ach or ACh, the last should be the best).

Answer: we have harmonized the format for this abbreviation for “ACh” throughout the text as suggested by the reviewer.